# Synergic versus Antagonist Effects of Rutin on Gallic Acid or Coumarin Incorporated into Chitosan Active Films: Impacts on Their Release Kinetics and Antioxidant Activity

**DOI:** 10.3390/antiox12111934

**Published:** 2023-10-30

**Authors:** Elizabeth Jacobs, Odile Chambin, Frédéric Debeaufort, Nasreddine Benbettaieb

**Affiliations:** 1Bioscience Department, Munster Technological University-Cork Campus, T12 P928 Cork, Ireland; 2Food and Wine Physico-Chemistry Unit, Institut Agro Dijon-Joint Unit Food Processing and Microbiology UMR PAM, Université de Bourgogne, 21000 Dijon, France; odile.chambin@u-bourgogne.fr (O.C.); nasreddine.benbettaieb@u-bourgogne.fr (N.B.); 3Department of Pharmaceutical Technology, UFR des Sciences de Santé, Université de Bourgogne, 21079 Dijon, France; 4Department of BioEngineering, IUT-Dijon-Auxerre, Université de Bourgogne, 20178 Dijon, France

**Keywords:** combined antioxidant effect, structure-reaction relation, diffusivity, DDPH, release

## Abstract

This work deals with the study of the release and antioxidant activity kinetics of three natural antioxidants associated as binary mixture (coumarin, and/or gallic acid and rutin) from chitosan films. Antioxidants were incorporated into film alone or in binary mixture. The aim was to determine the influence of rutin on the phenolic acid and benzopyrone. The UV-visible light transmission spectra of the films were also investigated. Neat chitosan films and chitosan incorporated coumarin exhibited high transmittance in the UV-visible light range, while GA-added chitosan films showed excellent UV light barrier properties. The molecular interactions between chitosan network and antioxidants were confirmed by FTIR where spectra displayed a shift of the amide-III peak. Rutin has a complex structure that can undergo ionization. The chitosan network structure induced change was found to influence the release behavior. The film containing rutin showed the highest antioxidant activity (65.58 ± 0.26%), followed by gallic acid (44.82 ± 3.73%), while coumarin displayed the lowest activity (27.27 ± 4.04%). The kinetic rate against DPPH-free radical of rutin is three times higher than coumarin. The kinetic rates were influenced by the structure and interactions of the antioxidants with chitosan. Rutin exhibited a slow release due to its molecular interactions with chitosan, while coumarin and gallic acid showed faster release. The diffusion coefficient of coumarin is 900 times higher than that of rutin. The rutin presence significantly delayed the release of the gallic acid and coumarin, suggesting an antagonistic effect. However, their presence weakly affects the release behavior of rutin.

## 1. Introduction

Nowadays, extending the lifespan of bioproducts and food through packaging has emerged as an important issue for the modern food industry. Extensive research efforts are currently directed towards the requirement for bio-based and/or edible films and coatings with excellent water and oxygen barrier properties to safeguard food, pharmaceuticals, and cosmetic bioproducts. Concurrently, there is a growing consumer demand for safe materials, particularly those derived from renewable sources such as agricultural and food industry waste and by-products [1].

Biopolymers have been widely used for the development of food packaging films as they are excellent vehicles for incorporating a wide variety of additives such as antioxidant, antifungal, and antimicrobial agents or neutraceutics [2]. Polysaccharides are often preferred instead of proteins. Indeed, the interactions with bioactive compounds are much more complex and more difficult to tailor for proteins because of the different types of bonds able to settle between protein amino acids and antioxidants [3].

Among biopolymers, chitosan is very promising for food packaging materials as it is nontoxic, biodegradable, and biocompatible [4]. Chitosan is obtained after alkyl deacetylation of chitin, which is derived from arthropod exoskeletons (e.g., crab, shrimp), fungal cell walls, etc., and is generally recognized as safe according to the U.S. Food and Drug Administration [5]. Chitosan is a cationic linear polysaccharide consisting of randomly distributed β-(1-4)-linked D-glucosamine (deacetylated unit) and N-acetyl-D-glucosamine (acetylated unit) [6]. Chitosan arouses growing interest for food packaging for its excellent physical properties (mechanical properties, transparency, etc.) and functionality (natural antibacterial and antioxidant activity). Recently, Souza et al. [7] displayed the great potential of chitosan for composite packaging materials, despite their high cost and relatively poor functional properties. They demonstrated that chitosan has more advantages than pristine petrol-based polymers for food packaging applications, particularly when used for active/intelligent materials. Indeed, chitosan coatings onto PLA films have been developed for shelf-life prolongation of fruits and vegetable [8]. Chitosan was very often used as a vehicle for active compounds such as essential oils, antioxidants, fruit extracts or pH indicator for bioactive or intelligent food packaging [9].

In the food industry, synthetic antioxidants like butylated hydroxyanisole (BHA), butylated hydroxytoluene (BHT), and tert-butylhydroquinone (TBHQ) have been commonly employed to inhibit lipid peroxidation. However, previous research has indicated that the accumulation of BHA and BHT in the body can lead to liver damage and increase the risk of carcinogenesis [10]. In recent years, there has been a growing focus on the exploration of natural bioactive compounds incorporated into food packaging and coatings to reduce the use of synthetic preservatives, primarily to answer the consumer demand. These bioactive compounds, which are predominantly non-toxic, are utilized as protective materials to preserve the quality of food and prolong its shelf life [2]. Therefore, herbs and spices are target sources of natural antioxidants; among these the phenolic compounds are widely distributed and possess the ability to scavenge free radicals [10]. These active compounds exhibit strong antioxidant properties by quenching free radicals, thereby inhibiting food oxidation. This is primarily due to their high capacity for hydrogen and single-electron exchange, which limits the transfer of oxygen and reduces the reactivity of free radicals. As a result, they provide protection against oxidation and extend the shelf life of packaged food products [11].

Coumarin is a phenolic compound naturally present in various plants, including cinnamon, cloves, Tonka beans, celery, and apricots. It possesses a unique chemical structure consisting of an aromatic ring fused to a condensed lactone ring and is known for its distinct fragrance, which can be described as spicy, fresh-hay, or vanilla-like. In addition to its aromatic properties, coumarin exhibits several bioactive characteristics, including anticoagulant, antimicrobial, antifungal, and antioxidant effects [12]. Interestingly, despite its bioactive properties, the utilization of coumarin in active packaging, as well as in edible films and coatings, has not been extensively investigated. Rutin is a well-known citrus flavonoid glycoside containing flavonol aglycone quercetin and disaccharide rutinose. It is commonly found in several plant sources, including buckwheat, tea, passionflower, apple, tobacco, forsythia, viola, and others. Beyond its presence in plants, rutin is also recognized as a valuable nutritional component and has applications in human medicine. The functional properties of rutin have been recognized as an efficient bioactive compound for making various biopolymer-based active packaging films [13]. Gallic acid is a prominent hydroxybenzoic acid compound that is widely distributed in various natural sources, including nuts, tea, oak bark, sumac extracts, and several other plants. It has been extensively studied for its diverse bioactive properties [14,15]. Gallic acid has been shown to exhibit antibacterial properties, which can help inhibit the growth and proliferation of bacteria and is an efficient antioxidant to neutralize free radicals and reduce oxidative stress in the body.

By incorporating coumarin and/or gallic acid, and rutin, into packaging materials or edible films/coatings, their potential as an active compound can be evaluated, offering new possibilities for enhancing food quality and extending shelf life. These mentioned naturals phenolic compounds are generally recognized as safe (GRAS) by the U.S. Food and Drug Administration and by the European Commission so they are considered to present no risk to the health of customers supporting their use in the development of active food packaging [16]. 

Aiming to prevent oxidation, particularly for ready-to-eat or sliced fatty products, a fast release just after packing is sought and then a slower and controlled release during storage is required to maintain a satisfactory antioxidant concentration at the surface. In such cases, the tune of the release mechanism is important and must be well tailored. In the case of long-term food storage, there is indeed a need for antioxidants that can exhibit both faster and slower release rates and actions. Antioxidants play a crucial role in preventing or slowing down the oxidation of fats, oils, and other components in food, thereby preserving its quality and extending its shelf life. However, the release rate and action of antioxidants need to be carefully controlled to ensure their effectiveness throughout the storage period. The critical point for the efficiency of the active system is to provide the best availability of the incorporated active compound to the food item in the exactly required period. This led to the design of special controlled delivery systems where hydrocolloids, especially chitosan, found their perfect place. Controlled release contributes to increasing the efficacy of active compounds on packed food surface during storage time and it requires the knowledge of both the controlled release mechanism and the reaction kinetics. This is the general rule for antioxidant agents.

The interaction between the phenolic compounds and the film matrix could affect their release and bioactivity. The film structure can act as a barrier, controlling the diffusion of the compounds and modulating their release kinetics. Interactions such as hydrogen bonding, electrostatic interactions, or physical entrapment can affect the release behavior and stability of the compounds [17].

The design of an active packaging with antioxidant properties would allow having an efficient system of controlled release during storage. On one hand, a faster release rate and action of antioxidants can be beneficial during the initial stages of storage or when the food is exposed to conditions that accelerate oxidation, such as high temperature or light. Indeed, is it well known that oxidation is greatly accelerated by the temperature and/or light on some foods like fish-based products [18], on meat-based products [19], on fatty products [20], and many other kinds of food [21,22]. These antioxidants are designed to rapidly scavenge free radicals and inhibit oxidation, providing immediate protection to the food. They act as a frontline defense, preventing the formation of off-flavors, rancidity, and nutriment degradation. On the other hand, a slower release rate and slow reactivity of antioxidants are desirable for long-term storage scenarios. They act as a long-lasting barrier against oxidative deterioration, ensuring the quality and safety of the food for an extended duration.

The antioxidant power of phenolic or benzenic compounds and/or flavonols has been investigated in some works, as well as their release from films into food simulants [23]. However, in our knowledge, no study focusses on their behaviors when they are mixed together in the film structure, which is closer to the real case of application in order to understand their interaction. The combination of phenolic compounds can result in synergistic or antagonistic effects on their release profile and bioactivity. Synergistic effects occur when the combined action of the compounds enhances their individual activities, resulting in a more pronounced effect. Antagonistic effects happen when the presence of one compound diminishes or interferes with the activity of the other. Using only one antioxidant in active packaging cannot provide both a faster and slower release profile. Consequently, the combination of antioxidants with different properties can achieve the desired release rates and actions (both fast acting and slower release). 

The originality of this work consists of displaying how a mixture of two phenolic compounds could affect their release profile and the bioactivity of each in relation to the film structure. Indeed, very few publications on active packaging film having antioxidant properties consider the synergy or competition between several compounds. Combining the rutin (higher MW, higher number of hydroxyl groups, and higher antioxidant activity) with gallic acid or coumarin (both with a medium MW and lower activity) allowed evaluating the effect of this molecule on the phenolic acid and benzopyrone behaviors in mixture in the chitosan matrix. Therefore, this work focused on the antioxidant activity (DPPH assay) and the release kinetics of phenolic compounds (rutin, gallic acid, and coumarin) alone or in a binary mixture, from chitosan films into aqueous food simulant and their radical scavenging activity. Specific attention was paid to their intermolecular interaction with the chitosan matrix.

## 2. Materials and Methods

### 2.1. Materials

Chitosan of commercial grade (France Chitine, MW = 165 kDa, low viscosity, 85%, deacetylation degree, Orange, France) was used as the matrix to form a continuous and cohesive network in the film. Anhydrous glycerol (Fluka Chemical, 98% purity, Buchs, Switzerland) plays the role of a plasticizer during film making. The coumarin (Coum) (1,2-Benzopyrone, purity ˃ 98%), the gallic acid (GA) (3,4,5-Trihydroxybenzoic acid, purity ˃ 98%), which are both simple phenolic compounds, and the rutin (Rut) (3,3′,4′,5,7-Pentahydroxyflavone 3-rutinoside, purity > 98%), which is a flavonol, were all purchased from Sigma-Aldrich (Saint-Quentin Fallavier, France). Only the commercial name of the antioxidants will be used in the manuscript and not the IUPAC chemical denomination. These were chosen as model of natural antioxidant molecules. Because most of this kind of antioxidants have the same absorption wavelength, the main reason to choose rutin, gallic acid, and coumarin is their ability to be discriminated by UV-Vis spectroscopy when in mixture. More sophisticated methods like HPLC, etc., could also be suitable but requires a very long time to assess kinetics and are almost not compatible with fast release. So, the main reason for our choice was the technical feasibility at first. The chemical structure and their physical-chemical parameters are given in Appendix A. 

### 2.2. Active Film Preparation

First, 60 g chitosan (CH) powder was dispersed in 3 L acetic acid solution at 1% (*v*/*v*) using a high shear homogenizer (Ultra Turrax, RW16 basic-IKA-WERKE, Staufen, Germany) at 700 rpm and at 25 °C. When the solution was completely homogenized, 6.6 g of glycerol (10%, *w*/*w* dry matter of chitosan) was added to this film-forming solution (FFS), under stirring (300 rpm). The chitosan solution had a pH value of about 4.8–5. Then, 400 mL of this FFS was kept as control (chitosan FFS). Then, 0.4 g of coumarin (Coum) or rutin (Rut) or gallic acid (GA) was separately added to another 400 mL of chitosan FFS aliquots under stirring to obtain active FFS at 5% (*w*/*w*) (antioxidant/dry matter of chitosan). To prepare the active FFS with mixture of rutin + coumarin or of rutin + gallic acid, 0.4 g of each compound was incorporated into 400 mL of chitosan FFS in the same conditions as previously described. 

The concentration of 5% of antioxidants (*w*/*w* of dry matter of chitosan) was chosen according to the literature. Indeed, most of the active films contained from 1% to 5% antioxidants or antimicrobials. When the active FFS were completely dissolved, an aliquot of 30 mL of each FFS was then poured into square plastic Petri dishes (12.5 × 12.5 cm^2^). The aqueous FFS solvent was removed by evaporation in a ventilated climatic chamber (KBF 240 Binder, ODIL, Dijon, France) at 25 °C and 50% RH for 18 to 24 h. After drying, the films were peeled off from the surface and stored up to weight equilibrium in a ventilated climatic chamber (KBF 240 Binder, ODIL, France) at 50% RH and 25 °C before analysis. All film formulations have been coded as described in Table 1.

### 2.3. Film Thickness Measurement

The film thickness was measured with an electronic gauge (PosiTector 6000, DeFelsko Corporation, Ogdensburg, NY, USA). Five replicates were done for each film sample, one from the center and four along the perimeter. The mean value was used in further calculations and standard deviation considered for the relative error.

### 2.4. UV-Visible Spectroscopy

UV-Vis absorption spectra of film samples were recorded in the wavelength range of 200 to 800 nm at one cycle, with a medium scan speed (200 nm.min^−1^) using a UV-Vis scanning spectrophotometer (Uvikon UV-VIS XS, Secoman with Model LabPower software V91640, Ales, France). Rectangular pieces (~2 × 5 cm^2^) of each film sample have been cut and stored in the test cell. The air was considered as the reference (empty cuvette). The results have been expressed as percentage transmittance (T%).

The opacity degree (mm^−1^) of different films was estimated according to Equation (1) at 260 nm for the UV-C range and at 380 nm for the UV-A range and at 600 nm for visible light:(1)$$Opacity\;degree=\frac{A_{\lambda }}{d}$$
where, *A_λ_* is the absorbance of a rectangular film sample with an incidence wavelength of 260 or 380 or 600 nm and d is the film sample thickness (mm).

### 2.5. Fourier Transform Infrared Analysis (FTIR)

Fourier-transform infrared spectroscopy (FTIR) of different formulations were acquired using an IR spectrophotometer (Spectrum 65; Perkin-Elmer, Haguenau, Saint-Quentin Fallavier, France) equipped with an Attenuated Total Reflectance (ATR) attachment with a ZnSe crystal. The face of the film exposed to air during the film drying process is placed into contact with the ZnSe crystal and measurements were done at room temperature (25 °C), using a wavenumber range from 4000 to 600 cm^−1^, at a resolution of 2 cm^−1^, and 32 scans. The results were represented as transmittance and the data treated using the Spectrum Suite software. Duplicate measurements were done for every sample. The incorporation of antioxidants into the polymer chains induced changes at a molecular scale that have been identified by the data analysis. Only the shifts of peaks of interest were considered to discriminate the chemical group interacting between chitosan and antioxidants and among antioxidants themselves.

### 2.6. Antioxidant Efficacy Testing (DPPH Test)

The Radical Scavenging Activity or Antioxidant Activity (RSA or *AA*, %) of films was measured using the 2,2-diphenyl-1-picrylhydrazyl (DPPH%), a stable free radical. Then, 6 cm^2^ of films (whatever the formulation) were introduced into a glass vial containing 10 mL DPPH solution (50 mg.L^−1^) in ethanol (96%, *v*/*v*). Glass vials, and consequently the contained solution, were isolated from light by covering with an aluminum foil. This ratio film surface/volume of solution was chosen to be similar to the ratio used for the release kinetic (6 dm^2^ of film in 1 L of liquid food simulant), which corresponds to the migration test according to European Regulation 10/2011 and 1416/2016 [16,24].

The disappearance of the DPPH• reactant (free radical) with time was assessed by measuring the absorbance at 515 nm using Jenway 6305 UV-Visible spectrophotometer (Thermo Fisher Scientific, Waltham, MA, USA). Furthermore, 2 mL of the solution were taken periodically to follow the kinetic; once the absorbance had been measured, the sample was returned to the flask to maintain the same volume of reaction medium. Samples were kept under agitation (300 rpm) in sealed flasks until the end of the experiment (equilibrium). The % DPPH inhibition is used to calculate *AA* antioxidant activity (or radical scavenging activity) according to the following equation (Equation (2)):(2)$$AA\left(t\right)=\frac{A_{blank\;(t)}-A_{sample\;(t)}}{A_{blank\;(t)}}\;\times \;100$$
where $A_{blank\;(t)}$ is the absorbance of DPPH solution without the film and $A_{sample\;(t)}$ is the absorbance of DPPH solution containing the film sample, both at time *t*. This test was carried out in triplicate.

The *AA* (%) at the end of the kinetic (at equilibrium) was discussed. The time to reach the equilibrium and the time to reach half (t_AA50%_) and 90% (t_AA90%_) of antioxidant activity (*AA*50% and *AA*90%) was also discussed. Due to absence of the lag phase in the *AA* (%) during the time, the kinetic rate constant (%.min^−1^) was determined as the slope of linear part of the curve (AA vs. time until the t_AA50%_) and discussed as the rate of DPPH free radical scavenging.

### 2.7. Release Kinetics of Antioxidants from Active Film into Food Simulant

The kinetic release of antioxidant from active chitosan films was assessed by immersion of films in food simulant (both side of film immersed), according to European Regulation 10/2011. Distilled water (pH = 6.5) was used as non-acidic aqueous food simulant assigned for foods that have a hydrophilic character and are able to extract hydrophilic substances [25].

For this test, pieces of film (60 cm^2^) were immersed into 100 mL of simulant at 25 °C. To prevent the effect of boundary/stagnant layers at the film interface, even in case of non-viscous liquids, a dissolution apparatus type II equipped with small paddles (Vankel, Livonia, MI USA) was used to maintain stirring at 150 rpm for the whole experiment time. This film surface/simulant volume ratio (60 cm^2^/100 mL) was chosen according to the standard test for measuring migration [24,26]. Around 2 mL of simulant was periodically sampled until equilibrium, and the OD was measured with a Uvikon XS UV-Visible spectrophotometer (Uvikon UV-VIS XS, Secoman, Ales, France) at corresponding maximum wavelength (*λ_max_*). The *λ_max_* (316, 256, and 356 nm for coumarin, gallic acid, and rutin, respectively) was previously determined from a complete UV-spectra of the pure antioxidants in food simulants (20 mg L^−1^). Prior experiments allowed selecting the wavelength specific of each compound without interferences to be sure that there was separate absorbance for each compound. Released antioxidant was quantified by UV spectroscopy. Calibration solutions were prepared, from 0 up to 40 mg L^−1^ of pure antioxidant (mg) by liter of food simulant for the calibration and determination of the Beer–Lambert’s law linear coefficient. The experiments were carried out in triplicate. To assess the synergistic or antagonistic effects of antioxidant on their release profile and kinetics parameters, the tests were done for each antioxidant incorporated alone into chitosan films and for both mixture rutin + gallic acid or rutin + coumarin.

The effective diffusion (*D*, m^2^/s) and partition (*Kp*) coefficients are two parameters commonly used to describe the release behavior of an active compound from a polymer film into a liquid food or food simulant. Diffusivity indicates how fast the active compound moves within the film, and partition coefficient indicates how much the active compound has affinity for the film or for the food simulant at equilibrium [27]. Partition coefficient ($K_{F/S}=\frac{C_{F,\infty }}{C_{S,\infty }}$: as the ratio of active concentration, at equilibrium, between film and simulant) was calculated when the equilibrium was reached. On the contrary, the effective diffusion coefficient of antioxidant in the film (*D*) was estimated from the analytical solution (integration) of the 2nd Fick’s law, according to the release kinetics data, and considering the transient state of the transfer with some assumptions:-At t = 0, the antioxidant is uniformly distributed throughout the films and its concentration in the film (*C_F_*) is uniform and equals the initial quantity (*C*_0_) incorporated in film during the film forming process, and the concentration of antioxidant in food simulant (*C_S_*) is considered equal to zero;-The diffusivity is constant and depends only on the temperature or external factor;-The concentration of antioxidant is uniformly dissolved in the simulant; the antioxidant is homogeneously dispersed in the simulant as it is continuously stirred;-There is no boundary layer resistance on the mass transfer process, and no interaction between the polymer (packaging) and the food simulant occurred.

The model used to integrate the 2nd Fick’s, which is close to reality, considers a packaging film material of finite dimensions, in contact on one side with food or food simulant. The model previously used by Benbettaieb et al. [27] or by Fu and Kao [28] is not often used to estimate the release of antioxidants since the ratio of the volume of food by packaging is very large (larger than 20–50) [29]. The mathematical model developed by Crank in 1975 (Equation (3)) with a finite volume of packaging and a very large volume of food or simulant is most appropriate in our case.
(3)$$\frac{C_{S,t}}{C_{S,\infty }}=1-\sum_{n=0}^{\infty }\frac{8}{{\left(2n+1\right)}^{2}.\pi^{2}}\exp\left(-\frac{D{\left(2n+1\right)}^{2}.\pi^{2}.\;t}{4L^{2}}\right)$$
where $C_{S,t}$ and $C_{S,\infty }$ are the concentrations of antioxidant (mg/L) in the food simulant at time t and at equilibrium, respectively. *D* is the effective coefficient (m^2^ s^−1^) and *L* (m) is the half of total thickness of the film (as the diffusion occurred from both sides of films).

Equation (3) was applied to the experimental release kinetics (up to equilibrium) and the model fitted by minimizing the sum of the square of the differences between measured and predicted values, using the Levenberg–Marquardt algorithm, and taking *D* as an adjustable parameter, with *n*-value: [0 1000]. Modeling was performed using Matlab/Simulink software environment (Matlab V8.5-2015, Mathworks, Natick, MA, USA). The mean value of effective diffusion was calculated from the triplicate analyses of each formulation. The coefficient of determination (R^2^) and the Root Mean Square Error (RMSE, mg/L) were determined from each measure.

### 2.8. Statistical Methods

SPSS 13.0 software (Stat-Packets Statistical analysis Software, SPSS Inc., Chicago, IL, USA) was used for the statistical analysis of data. The significant difference between values were assessed through the multiple comparisons of means from the analysis of variance (one-way-ANOVA test). The LSD (Least Significant Difference) mean comparison test was used at the significance level of 95% (*p*-value < 0.05) to compare all the parameters analyzed between different formulations. A student *t*-test was used for comparisons between two groups of data (simple comparison of two means) at the same confidence level (95%).

## 3. Results and Discussions

The antioxidant efficiency of releasing systems, such as edible chitosan-based films and coatings, is strongly related to their ability to be released. Therefore, the interactions involved in the chitosan and antioxidants, and consequently the network structure changes, have been studied by FTIR. Indeed, the release parameter such as the apparent diffusivity of the active compounds in the film and their partition between the film and food simulant are film structure dependent. These data on the release behavior have to be related to the antioxidant activity. The sensitivity of the coated or packaged foods to be oxidized in presence of light or UV is also a crucial parameter to consider, that is why UV-visible light spectra of films have been analyzed.

Preliminary observation of the sample displayed color changes of films containing the antioxidants (Table 2). Coumarin is also known to have an odor impact; however, only the odor related of acetic acid was perceived during the two days after film making, and no further detectable odor has been smelled.

### 3.1. Optical Properties of Films

The UV-visible light transmission spectra of all the studied films are given in Figure 1. Table 2 displays the opacity values of different films calculated for UV-C at 260 nm, for UV-A at 380 nm, and for visible length at 600 nm. The film transparency is of great importance for food packaging marketing and/or should consider the light oxidation of some foods. It is the opposite of opacity that is commonly estimated from the transmission at 600 nm. In general, transparent films are preferred, but food spoilage could result from light or UV exposure and, consequently, barrier properties for light or UV could be needed [30].

The neat chitosan film displays the highest transmittance in both UV and visible light range, they are less colored and fully transparent as displayed in pictures (Table 2). Oppositely, the GA-added chitosan films (alone or mixed with rutin) had a 0% transmittance at a wavelength less than 360 nm, indicating an excellent barrier against UV light, but is also the more efficient in the visible range. As can be seen in Figure 1, gallic acid possess light- and UV-barrier properties due to its opacifying power and when dispersed in the chitosan network has the ability to diffract and scatter light rays. Thus, the design of light barrier chitosan films with GA or mixture GA/rutin induced bioactive films with excellent UV light barrier properties is promising for packing highly sensitive photo-oxidative foods (such as cheese and unsaturated fat food).

The films containing rutin alone or the mixture rutin and coumarin have the same UV-visible profile. The opacity values at 260, 380, and 600 nm were three-fold higher compared to chitosan films. On the contrary, films with coumarin alone had a higher light transmittance (and lower opacity values) in the visible range between 400 and 500 nm compared to the film containing the mixture of rutin and coumarin. Furthermore, the chitosan film containing the rutin or the mixture of rutin and coumarin exhibits two main absorption bands at around 260 nm and at 380 nm, referring to the (π)→(π*) transitions of aromatic rings as previously reported by Lipatova et al. [31]. The lower UV transmittance (and higher opacity) of these films compared to neat chitosan film was due to the strong UV absorption ability of rutin [2]. Rutin is known for its ability to absorb wavelengths between 400 and 600 nm, corresponding to the color blue, which explains the yellow-greenish coloration of films containing rutin [32].

### 3.2. Molecular Characterization of Films by Infrared Spectroscopy (FTIR)

The presence of new interactions or bonds and the nature of the linkage between the chitosan network and the antioxidants were firstly identified by Fourier transform infrared spectroscopy (FTIR). FTIR spectra of all films are displayed in Figure 2. The spectrum of control films (without antioxidant) showed the characteristic peaks of chitosan films as observed and identified by some authors [33,34].

The band located at 3000–3500 cm^−1^, centered at around 3361 cm^−1^, is assigned to νOH stretching of free water and νNH stretching of amide A. The amide A band (3000–3500 cm^−1^) overlaps the OH band. The shifts can, therefore, be attributed indiscriminately either to changes in water content or to water–biopolymer interactions. Consequently, the part of the spectrum in the 2500–4000 cm^−1^ region is not further discussed.

According to Mohan et al. [35], the νCH symmetric stretching of amide B was indicated by the bands obtained at 2880 cm^−1^. The peak located at 1655 cm^−1^ is assigned to C=C and C=O stretching of primary and secondary amine N-H band of amide I (acetyl group of chitosan) [13].

The band located at 1550–1610 cm^−1^, with centered peak at 1560 cm^−1^, is assigned to δNH of amide II, and the peak centered at 1340 cm^−1^ is assigned to aromatic primary amine, C-N stretch of amide III [1]. Otherwise, the CH bending is indicated by the bands obtained at 1410–1420 cm^−1^ (1412 cm^−1^ in our spectra) [35]. The same authors displayed that the C-O-C band stretching was responsible for the peaks at around 1153 and 1031 cm^−1^.

The 1074 cm^−1^ peak could be related to interactions settled between the plasticizer (OH group of glycerol) and the chitosan via hydrogen bands as previously identified by some authors [1].

No shift was detected in the amide B after the incorporation of antioxidant related to the νCH (2880 cm^−1^) peak position. Amide-I (1655 cm^−1^) and amide-II (1560 cm^−1^) peak positions were not significantly changed and as well as for the peak attributed to CH bending (1410–1420 cm^−1^). Similarly, Benbettaieb et al. [36] previously reported that there is no change observed in amide-I, amide-II, and amide-III peak positions for chitosan-gelatin films after incorporation of coumarin. On the contrary, Lipatova et al. [31] displayed that the amide-I (1633 cm^−1^) and amide-II (1546 cm^−1^) of chitosan film shifted to 1641 and 1550 cm^−1^, respectively, after rutin incorporation at 3% (*w*/*w*). They attributed these shifts to the electrostatic interfacial interaction and to hydrogen bonding between chitosan amino groups and rutin. Rutin contains electron-donating phenolic OH groups, which increase the electron density of the benzene rings. Rutin is, thus, highly susceptible to interactions with the positively charged NH_3_^+^ groups of protonated chitosan. These interactions are mainly due to electrostatic forces. Similarly, Narasagoudr et al. [2] reported an important shift in the band at around 1639 cm^−1^ to a higher wavenumber when rutin is added to chitosan and PVA films.

The amide-III peak (1340 cm^−1^, for chitosan film) displays a significant shift to 1355 cm^−1^ for films containing gallic acid or rutin and for films with the mixture rutin+coumarin or rutin+gallic acid. However, there is no shift in this peak position after incorporation of coumarin alone. At a pH around 4.8 to 5 (pH of film forming solutions), the shift on amide-III peak can be related to the electrostatic interaction between the charged amino group (NH_3_^+^) of chitosan and the charged carboxylic group (COO^−^) of the phenolic ring of gallic acid, for which the pKa is around 4 (pH ~ pKa + 1). For rutin, as it is a complex molecule with multiple functional groups, it does not have a single pKa value. However, different functional groups within rutin can have their own pKa values. The most significant functional groups in rutin are the hydroxyl groups (-OH) and the phenolic hydroxyl group (-OH) attached to the flavonol quercetin moiety. These groups can undergo ionization and influence the overall acidity or basicity of rutin and in the pH condition during chitosan preparation can interact with protonated amino group of chitosan.

The coumarin is a molecule typically uncharged or slightly negatively charged. Therefore, the yield of electrostatic interactions between chitosan and coumarin is lowered. Indeed, a dipole–charge interaction may occur, and to a lesser extent, a dipole–dipole interaction might occur. Furthermore, coumarin has only one hydrogen bond acceptor site (carbonyl group) and is without hydrogen bond donor site. Chitosan, with its amino and hydroxyl groups, can act both as a hydrogen bond donor and acceptor and can react with only the hydrogen bond acceptor site of coumarin. Therefore, there is no strong interaction or association between chitosan and coumarin and a part of this compound moved in the surface of chitosan films after the drying process.

As can be seen in Figure 2, there is no significant shift in peak position related to the C-O-C band stretching at around 1153 and 1031 cm^−1^ after antioxidant incorporation, contrarily to Narasagoudr et al. [2] that displayed an absorption peak at about 1032 cm^−1^ corresponding to the C-O group shifted to 1053 cm^−1^. These observations reveal that the interaction mainly occurred between hydroxyl and amino groups of chitosan and hydroxyl groups of rutin. From our FTIR spectra data, the peak observed at 1074 cm^−1^ in chitosan film is shifted to 1069 cm^−1^ after coumarin addition and to 1067 cm^−1^ after gallic acid or rutin or the mixture rutin+coumarin addition and to 1064 cm^−1^ after incorporation of the mixture rutin+gallic acid.

Thus, the groups involved in the interaction mechanisms were identified by FTIR. The release of phenolic compounds from the film into the aqueous medium is, therefore, likely to be affected by these interactions, as is their antioxidant activity.

### 3.3. Kinetics of Antioxidants Activity (from DPPH Test)

The antioxidant property of active films was assessed using the method based on the scavenging of the DPPH• radical molecule. Figure 3 displays the antioxidant activity kinetics (calculated from Equation (2)) in ethanol. The difference observed in the DPPH kinetics between these antioxidants should be related firstly to the difference on their structure and secondly to their release kinetics from the chitosan network to the DPPH media. The structure of phenolic compounds is a key determinant of their radical scavenging activity, and this is referred to as structure–activity relationships. The structure-activity relationships of flavonoids are generally more complicated than those of hydroxybenzoic and hydroxycinnamic acids due to the relative complexity of the flavonoid molecules. The degree of hydroxylation as well as the position of the hydroxyl groups increased the radical scavenging capacity of flavonoids.

Indeed, CH and CH-Coum films displayed the lowest *AA* activity (less than 30%), and the plateau was reached after 210 min, without any significant difference between the two formulations. Chitosan films have a natural scavenging activity. The same behavior was previously observed for films composed of a blend of chitosan and gelatin [1]. Antioxidant activity can be attributed firstly to the NH2 and COOH groups of gelatin or chitosan, which are primarily responsible for antioxidant activity, and secondly to the OH groups of chitosan, which have the ability to scavenge radicals. Roy and Rhim [13] confirmed that the antioxidant activity of chitosan films is related to the -OH (C_6_) and -NH2 (C_2_) groups of chitosan.

For the active chitosan films, phenolic compounds exhibited two phases in their interaction with the radical, the first phase being rapid (first 20 min) followed by a slower phase in the next 50 min to reach equilibrium, which is the case for most of the antioxidant behaviors against DPPH• radical (Figure 3 and Table 3). CH-Coum films have a very weak scavenging ability (27%) at the same level of chitosan films (29%). On the contrary, CH-GA films showed higher *AA* (44.8 ± 3.7%) compared to neat chitosan films or chitosan films with coumarin, but the plateau is reached at the same time (210 min). CH-Rut and CH-Rut+GA films showed the fastest kinetic and exhibited the highest AA leading up to a 65.5 ± 0.26% and 63.4 ± 1.15%, respectively, at equilibrium (100 min). This can be explained by a higher intrinsic antioxidant activity of the rutin.

Jing et al. [37], observed similar results. They used a 3D-QSAR (quantitative structure–activity relationship) model to understand the structure–activity relationships and, thus, to predict phenolic acids scavenging activity. They reported that the concentration of pure phenolic compounds to scavenge the DPPH• radical (IC50 = *AA*50%, μM) of GA is ~180 times lower than that observed for p-coumaric acid. The high DPPH▪ scavenging activities of gallic acid are often attributed to the multiple hydroxyl substitution pattern of the phenolic acid. The presence of electron donating or high electron density hydroxyl group at meta and para positions of the gallic acid explains its good scavenging ability [11]. The addition of coumarin delays the scavenging kinetic of DPPH radical of films containing rutin; the plateau is reached after 210 min, but the final *AA* (65.7 ± 2.8% at the end of kinetic (300 min) remains the same as CH-Rut and CH-Rut+GA films.

The time to reach 50% of maximum scavenging ability (t_AA50%_) (Table 3) is less than 43 min for CH-Rut and CH-Rut+GA films. It increases until 56.6 min for CH films and up to 68.3 and 66 min for CH-GA and CH-Rut+Coum films, respectively. The t_AA50%_ of films containing coumarin still have the higher t_AA50%_ (76.3 min). On the contrary, films incorporating rutin or mixture of rutin and gallic acid have the lowest t_AA50%_ (42.5 and 28.3 min, respectively). Active films having the higher *AA* (%), exhibited the lowest t-_AA50%._ The same trend was observed for *AA*90% where the films incorporating coumarin still have the highest t_AA90%_ and film incorporating rutin or a mixture of rutin and GA have the lowest t_AA90%_ (Table 3).

The ability to donate electrons or hydrogens via the -OH of the flavonol quercetin and the disaccharide rutinose plays an essential role in rutin’s free radical scavenging [13]. A similar antioxidant activity has been observed in the rutin-added gelatin and starch-based films [38,39]. Furthermore, for the DPPH test, the presence of OH group in phenol rings is an indicator for a higher scavenging activity against DPPH free radical. Coumarin with only six hydrogens has the lowest *AA* (%) followed by gallic acid with three OH groups and two hydrogen groups and one carboxylic acid group. Rutin is a flovonol with 10 hydroxyl groups and 20 hydrogen groups and has the highest scavenging ability.

Pernin et al. [40] displayed that the chemical structure of phenolic compound and the presence of delocalization structures are mostly responsible for the antioxidant activity. They calculated the effective concentration able to delay oxidation for 1 h (EC1h in µmol.L^−1^.h^−1^). The lower this value, the stronger the antioxidant is. From their work, the EC1h of gallic acid is 3.8 µmol.L^−1^.h^−1^ against 1.1 µmol.L^−1^.h^−1^ for rutin. Whereas p-coumarin acid derived from coumarin has the highest EC1h (28.5 µmol.L^−1^.h^−1^) and, therefore, the lowest AA. This result confirms our findings.

Due to absence of the lag phase in the antioxidant activity displayed in Figure 3, the kinetic rate constant (%.min^−1^) was determined from the slope of the linear part of the graph (*AA* vs. time until t_AA50%_) and discussed as the rate of DPPH• free radical scavenging. The CH-Rut+GA film exhibited the highest kinetic rate (1.02 ± 0.09%.min^−1^) flowed by the CH-Rut film (0.78 ± 0.14%.min^−1^), then followed by the CH-Rut+Coum film (0.52 ± 0.12%.min^−1^), and then by the CH-GA (0.39 ± 0.09%.min^−1^), and finally, by the CH and CH-Coum (0.23 ± 0.04 and 0.31 ± 0.07%.min^−1^), respectively. The kinetic rates are then same trend as the t_AA50%_ and t_AA90%_ confirmed by the EC1h values from the literature as discussed before [40].

The differences in the antioxidant reaction behaviors may not only be due to differences in the mechanism of action of the antioxidants. In fact, their release kinetic from the chitosan network should also be considered, as it can influence their interaction with the DPPH media. In reality, the kinetic constants are the result of a complex set of reactions occurring in multiphasic systems in which partitioning the compounds between the films and the DPPH solution must be taken into account.

### 3.4. Combined Effect of Antioxidants on the Release Kinetics and Parameters

This part of study focuses on the effect of mixing the two antioxidants on their release in a food simulant. The experimental points were expressed as the ratio between the released antioxidant during the time and the antioxidant released at equilibrium (*Ct/C∞* vs. time) to normalize the data. Diffusion (*D*) and partition ($K_{F/S}=\frac{C_{F,\infty }}{C_{S,\infty }}$) coefficients were determined from the data plotted in Figure 4. As the concentrations of antioxidant involved are always much lower than their solubility in water, no concentration nor partition coefficient affected the diffusivity. According to Figure 4a–c, the experimental data showed an acceptable fit to the Fickian diffusion model (R^2^ > 0.88 and RSME < 5.3 mg/L). Figure 4a–c and Appendix A display the data of the release of Coum, GA, or Rut, and their mixture (Coum+Rut or Coum+GA) from the films into the aqueous food simulant.

The coumarin releases more rapidly, followed by the gallic acid, when rutin displays a very slow release (Figure 4a). For coumarin, the time at 50 and 90% released is 0.58 ± 0.14 and 1.26 ± 0.64 min, respectively (Table 4). These times increased, respectively, by 10 to 25 times in the case of GA and by more than 50-fold for rutin. The same trend was observed for the diffusion coefficient. The rate of release of GA and rutin in water was, respectively, 81 and 900 times lower than that of coumarin. The time to reach 50 and 90% of equilibrium and the rate of release (*D*, m^2^/s) were more related to the chemical structure and the physicochemical properties of antioxidant (molar volume, solubility in water). Rutin has the lowest release rate and the highest time to reach equilibrium due the higher molecular weight, molar volume in comparison with coumarin and GA, and lower solubility in water, slowing down the diffusion ability and its greater ability to establish interactions with the chitosan. The latter explained why the percentage of release of the rutin (8.4 ± 0.7%) is much lower (Table 4).

Regarding the partition coefficient, the $K_{F/S}$ values for rutin decreased by 48 and 75%, respectively, compared to coumarin and GA. The percentage of release and the partition coefficient were more related to the solubility and to molecular interaction between the antioxidant and the chitosan network as discussed from FTIR data. Indeed, rutin has the lowest solubility in water (0.15 g.L^−1^) compared to gallic acid, which is 100 time more soluble in water. Therefore, the release of rutin is hindered by its molecular interaction with charged amino group in chitosan leading to slower release kinetics. As rutin is a larger molecule, it presented more sites of interaction with chitosan backbone and so it influenced the network obtained during the manufacturing process and then the subsequent properties such as release.

Figure 4b shows the effect of the combination of two antioxidants on their respective release behaviors. The release of coumarin from the CH-Rut+Coum films is delayed by the presence of the rutin. Indeed, the diffusion coefficient of coumarin is reduced by 163 times when in mixture with rutin. The time to reach 50 and 90% of equilibrium also increased up to 55 times. Inversely, the presence of coumarin in CH-Rut+Coum films accelerates the release of the rutin. It can be supposed that coumarin disturbed the settlement of interactions between rutin and chitosan, giving a less organized network compared to only rutin added in chitosan solution. Therefore, an antagonist effect is observed on the coumarin behavior when rutin is added.

From Figure 4c, the rutin also delayed the release of the GA in the CH-Rut+GA films compared to CH-GA films. However, the presence of GA did not affect the release rate and the time to reach equilibrium of the rutin in the CH-Rut+GA films. In the case of GA and rutin mixture, none of the antioxidant affect the other; rutin interacted significantly only with the chitosan chains.

## 4. Conclusions

The study highlights the importance of the release efficiency of antioxidants from chitosan-based films and coatings. The interactions between chitosan and antioxidants were analyzed using FTIR, and the network structure was found to influence the release behavior. The UV-visible light transmission spectra of the films were investigated. Neat chitosan films exhibited high transmittance in both UV and visible light ranges, while GA-added chitosan films showed excellent UV light barrier properties and opacity at 600 nm. FTIR displayed the interactions between chitosan and antioxidants, particularly with the rutin, mainly due to the hydroxyl and amino group. The antioxidant activity of chitosan films containing different antioxidants was evaluated using DPPH radical scavenging. Rutin showed the highest antioxidant activity, followed by gallic acid, while coumarin displayed the lowest activity. The kinetic rates of antioxidant scavenging were influenced by the structure and interactions of the antioxidants with chitosan. The release kinetics of antioxidants from the films were investigated. Rutin exhibited a slow release due to its molecular interactions with chitosan, while coumarin and gallic acid showed rapid release. The presence of rutin delayed the release of coumarin, suggesting an antagonistic effect. However, the presence of gallic acid did not significantly affect the release behavior of rutin.

In summary, this study demonstrates the importance of understanding the interactions between chitosan and antioxidants to optimize the release efficiency and antioxidant activity of chitosan-based films and coatings. This system displays the ability of a fast release of one of the antioxidants for an immediate effect on the food, followed by a slower release of the rutin, which nevertheless has a greater antioxidant efficacy. This study provides valuable insights into designing packaging materials for photo-oxidatively sensitive foods and highlights the potential of incorporating different antioxidants to modulate the release kinetics and activity of bioactive films. The release efficiency of antioxidants in simulant can be translated into practical applications. Indeed, numerous scientific investigations during the last two decades have demonstrated that release from food contact materials into food or food simulants follow the same predictable physical processes, which take in account the absence of specific interactions with food. This system could be relevant for sliced foods applications like cheese, fresh fruits, or lightly processed fruits and vegetables whose surface oxidizes very rapidly after cutting and packing and then oxidation continues during storage. In this case, a fast release of antioxidant is required and followed by a slow release during a long time. The design of active packaging able to display this double effect is sought as we observed with our combination of two different type of natural antioxidants.

## Figures and Tables

**Figure 1 antioxidants-12-01934-f001:**
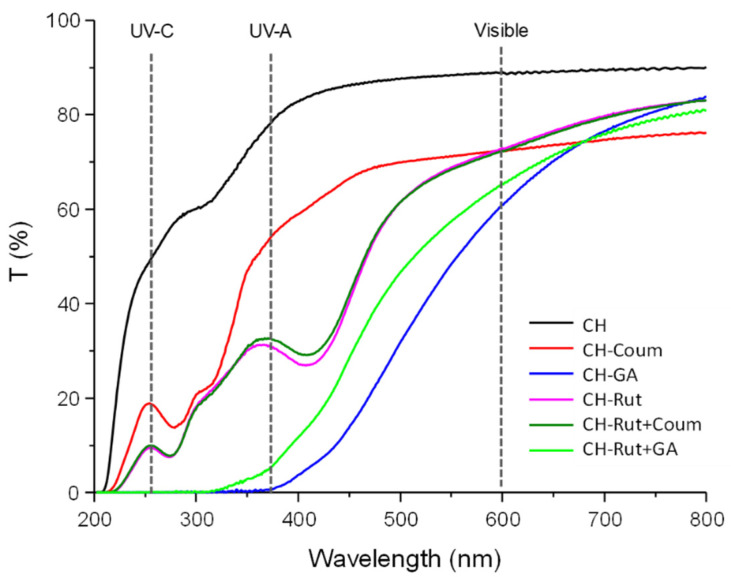
Transmittance (%) as a function of wavelength (nm) for all the tested films, a focus on the opacity degree at 260 nm for UV-C, at 380 nm for UV-A, and at 600 nm for visible light.

**Figure 2 antioxidants-12-01934-f002:**
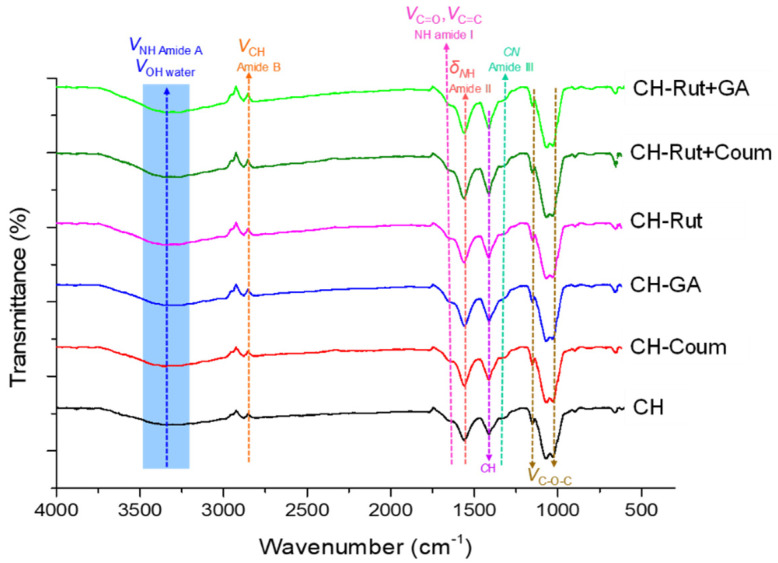
FTIR spectra as a function of wavenumber (cm^−1^) of different formulations.

**Figure 3 antioxidants-12-01934-f003:**
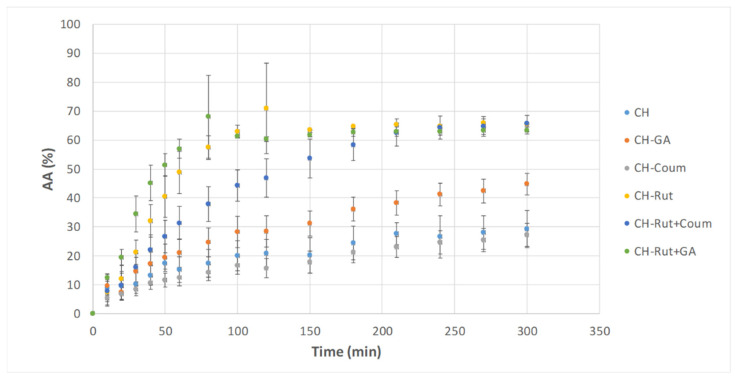
Kinetics of antioxidant activity of the different films containing antioxidant alone or binary blend.

**Figure 4 antioxidants-12-01934-f004:**
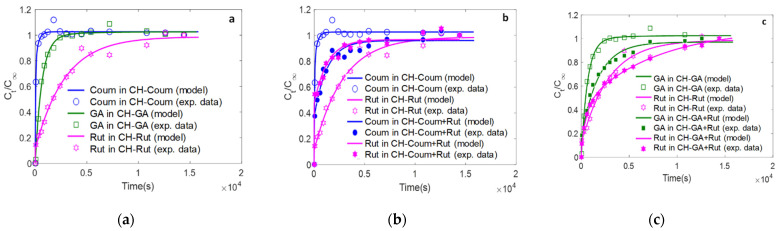
Release kinetics of antioxidants from active chitosan films into aqueous food simulant (distilled water) at 25 °C. (**a**) Release of coumarin from CH-Coum, of GA from CH-GA and of rutin from CH-Rut films; (**b**) release of coumarin from CH-Coum and from CH-Coum+Rut, of rutin from CH-Rut and from CH-Coum+Rut; (**c**) release of gallic acid from CH-GA and from CHGA+Rut, of Rutin from CH-Rut and from CH- GA+Rut. (Ct, concentration of antioxidants released in the food simulant at time t; C∞, concentration of antioxidants released in the food simulant at equilibrium. The lines are the fitting of the experimental values (symbols) from release kinetics data).

**Table 1 antioxidants-12-01934-t001:** Film sample code and description.

Film Sample Code	Description
CH	Chitosan films (prepared by casting method)
CH-Coum	Chitosan film incorporated coumarin at 5% (*w*/*w*)
CH-GA	Chitosan film incorporated gallic acid at 5% (*w*/*w*)
CH-Rut	Chitosan film incorporated rutin at 5% (*w*/*w*)
CH-Rut+Coum	Chitosan film incorporated a mixture of rutin and coumarin at 5% (*w*/*w*) each
CH-Rut+GA	Chitosan film incorporated a mixture of rutin and gallic acid at 5% (*w*/*w*) each

**Table 2 antioxidants-12-01934-t002:** Opacity degree to UV and light, and visual aspects of films.

Formulation	Degree of Opacity			Film Picture
	UV-C at 260 nm	UV-A at 380 nm	Visible Light at 600 nm	
CH	2.92 ± 0.30 ^a^	0.97 ± 0.91 ^a^	0.5 ± 0.06 ^a^	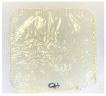
CH-Coum	7.44 ± 0.75 ^b^	2.55 ± 0.21 ^b^	1.39 ± 0.12 ^a,b^	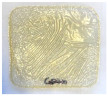
CH-GA	35.48 ± 0.33 ^d^	19.32 ± 2.03 ^e^	2.14 ± 0.19 ^c^	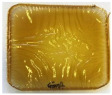
CH-Rut+Coum	10.17 ± 0.97 ^c^	4.94 ± 0.55 ^c^	1.39 ± 0.11 ^a,b^	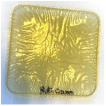
CH-Rut	10.39 ± 0.89 ^c^	5.21 ± 0.53 ^c^	1.36 ± 0.13 ^b^	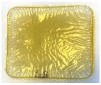
CH-Rut+GA	32.74 ± 0.31 ^d^	11.65 ± 1.10 ^d^	1.84 ± 0.16 ^c^	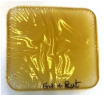

^a,b,c,d^ Mean with the same superscript letter in the same column are not significantly different at *p* ˂ 0.05.

**Table 3 antioxidants-12-01934-t003:** Maximum antioxidant activity at equilibrium (*AA*, %), the time (t_AA50%_ and t_AA90%_) (min) to reach 50% and 90% of maximum AA and the initial kinetic rate (%.min^−1^).

Formulation	*AA* (%) at Equilibrium	t_AA50%_ (min)	t_AA90%_ (min)	Kinetic Rate (%.min^−1^) *
CH	29.28 ± 6.34 ^a^	56.6 ± 5.7 ^c^	200 ± 17.3 ^c^	0.31 ± 0.075 (R^2^ ˃ 0.89) ^a,b^
CH-Coum	27.27 ± 4.04 ^a^	73.3 ± 11.5 ^b^	243.3 ± 15.2 ^d^	0.23 ± 0.047 (R^2^ ˃ 0.95) ^a^
CH-GA	44.82 ± 3.73 ^b^	68.3 ± 17.5 ^c,b^	225 ± 15 ^c,d^	0.39 ± 0.091 (R^2^ ˃ 0.93) ^b^
CH-Rut+Coum	65.71 ± 2.8 ^c^	66 ± 15 ^c,b^	180 ± 30 ^c^	0.52 ± 0.12 (R^2^ ˃ 0.98) ^c^
CH-Rut	65.58 ± 0.26 ^c^	42.5 ± 7.5 ^b^	91.6 ± 11.9 ^b^	0.78 ± 0.14 (R^2^ ˃ 0.98) ^d^
CH-Rut+GA	63.38 ± 1.15 ^c^	28.3 ± 3.5 ^a^	59.3 ± 4.6 ^a^	1.02 ± 0.094 (R^2^ ˃ 0.95) ^e^

Values are given as mean ± standard deviation. ^a,b,c,d,e^ Mean with the same superscript letter in the same column are not significantly different at *p* ˂ 0.05. R^2^ is the coefficient of determination corresponding to the kinetic rate determination from the first phase of *AA* kinetic. * Determined as the slope of first phase of the kinetics of *AA* (%) vs. time (min).

**Table 4 antioxidants-12-01934-t004:** Release parameters of antioxidants alone or in mixing in chitosan films (*n* = 3).

Formulation	Percentage of Release at the Equilibrium (%)	Time at 50% Released (min)	Time at 90% Released (min)	Diffusion Coefficient * (10^−15^ m^2^/s) * (R^2^/RMSE (mg/L)	Partition Coefficient $K_{F/S}=\frac{C_{F,\infty }}{C_{S,\infty }}$
Coum alone in film CH-Coum	15.6 ± 4.1 ^d^	0.58 ± 0.14 ^a^	1.26 ± 0.64 ^a^	30,830 ± 36,650 ^a^ (0.92/5.3)	11,753 ± 3157 ^a^
GA alone in film CH-GA	27.4 ± 3.4 ^e^	5.00 ± 1.73 ^b^	31.00 ± 3.60 ^a^	378 ± 280 ^d^ (0.92/3.1)	5584 ± 884 ^a^
Rut alone in film CH-Rut	8.4 ± 0.7 ^c^	25.01 ± 8.66 ^d^	87.66 ± 11.67 ^c,d^	36 ± 3 ^a^ (0.89/1.7)	22,622 ± 2147 ^c^
Coum in film CH-Coum+Rut	1.8 ± 0.3 ^a^	0.83 ± 0.28 ^a^	70.04 ± 17.32 ^c^	189 ± 119 ^b,c,d^ (0.95/2.3)	114,386 ± 26,963 ^e^
Rut in film CH-Coum+Rut	1.9 ± 0.2 ^a^	0.83 ± 0.15 ^a^	46.66 ± 5.77 ^b^	167 ± 18 ^c,d^ (0.98/1.8)	108,099 ± 14,703 ^e^
GA in film CH-GA+Rut	12.5 ± 2.6 ^d^	13.01 ± 2.64 ^c^	95.02 ± 5.02 ^d^	76 ± 9 ^b^ (0.88/1.5)	15,032 ± 3438 ^b^
Rut in film CH-GA+Rut	4.4 ± 0.5 ^b^	23.30 ± 5.77 ^d^	133.33 ± 41.63 ^e^	29 ± 5 ^a^ (0.97/1.9)	45,499 ± 5795 ^d^

Values are given as mean ± standard deviation. ^a,b,c,d,e^ Values with the same superscript letter in the same column are not significantly different at *p* ˂ 0.05. R^2^ is the coefficient of determination corresponding to the modeling of kinetics of release. RMSE is the Root Mean Square Error corresponding to the difference between the modeling values and the experimental data of kinetics of release and expressed on mg/L. *** Determined from modeling using Equation (3).

## Data Availability

The data presented in this study are available on request from the corresponding author or from author N. Benbettaieb.

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
