# Peer review of "Synergic versus Antagonist Effects of Rutin on Gallic Acid or Coumarin Incorporated into Chitosan Active Films: Impacts on Their Release Kinetics and Antioxidant Activity"

_antioxidants, 2023, doi:10.3390/antiox12111934_

Round 1
Reviewer 1 Report
Comments and Suggestions for Authors
The authors presented an interesting study. It is very interesting to consider synergistic or antagonistic interactions of antioxidants in polymer systems. Nevertheless, several issues should be adressed.
1. Line 25 - gelatin?
2. Does the citation style comply with editorial requirements?
3. An interesting observation could also be the analysis of fragrance release (coumarin has a characteristic smell) - did the authors note differences in the smell of the films and differences in mixed systems?
4. Due to the different mechanism of action and interactions with hydrophobic and hydrophilic antioxidants of DPPH and ABTS radicals, it is suggested that the study be expanded to include the ABTS radical.
5. Section 2.2. - please use space between numbers and units
6. Please revise manuscript for formatting - superscipts, "et al." instead of "and al.", etc. (line 246, etc.)
Reviewer 2 Report
Comments and Suggestions for Authors
There are a variety of natural compounds from plants that have antioxidant and antibacterial properties. Please explain the reasons for choosing to add coumarin, gallic acid, and rutin to packaging materials.
Please explain the reasons for these experimental formulations and the reasons for not having CH-GA+Coum(gallic acid and coumarin)formulation
Based on the release kinetics of antioxidants from active chitosan films into distilled water, does it demonstrate positive effects for food with high fat content?
Please explain Why does the AA(%) of CH-Rut in Figure 3 rapidly decline at 150 minutes?
Line 121 -124, please provide additional information on the effects of temperatures and light stimulation on food oxidation
In line 540. It should be Figure 4 not Figure 2, need minor corrections.
In line 594, it is mentioned the importance of release efficiency of antioxidants from chitosan-based films and coatings , does it can translate into practical applications, such as food preservation or biomedical applications
In Figure 1, please explain the sudden decrease of the transmittance of CH-Rut+Coum and CH-Rut at 400-450 wavelength (nm) .
Kindly furnish details regarding strategies for optimizing films comprising a variety of antioxidants listed in Table 3, with the aim of enhancing their efficacy and achieving visually appealing properties post-irradiation.
Kindly elucidate practical applications of thin films through illustrative examples.
Reviewer 3 Report
Comments and Suggestions for Authors
The article, entitled 'Synergic versus antagonist effects of rutin on two phenolic acids incorporated into chitosan active films: impacts on their release kinetics and antioxidant activity', concerns the study of the release kinetics and antioxidant activity of three natural antioxidants (coumarin, rutin and gallic acid) from chitosan films.
At the outset I would like to point out that the subject matter is very interesting, timely and I took the mentioned article for review with great enthusiasm, as the topics are very interesting, current and developing.
At the outset, I would like to point out that the authors have not very carefully analysed the literature, which in my opinion is poor (30 items). Admittedly, almost half are very up-to-date (articles dated 2020 or later). Some, on the other hand, are in my opinion not entirely cited on the spot. I will mention, for example, item 3 cited in lines 52-54.
However, one by one:
The keywords "partition and matrix network" in my opinion are not justified.
Line 48-49: 'Among the biopolymers, chitosan is the most interesting for food packaging as it is nontoxic, biodegradable and biocompatible'. A number of other polysaccharides are also characterised by the three properties mentioned. In order to justify this, stronger arguments would have to be made and additional literature would have to be used.
Section 2.1 reads "The coumarin (Coum) (hydroxycinnamic acid, purity ˃ 98%), the gallic acid (GA) (hydroxybenzoic acid, purity ˃ 98%), both are phenolic acids,.....). This is incorrect information. Coumarin is a compound of the benzopyrone group, an o-hydroxycinnamic acid lactone, not a carboxylic acid. Also incorrect are the given names coumarin and gallic acid. The correct names appear in the table. Speaking of the table, I think the table is completely out of place. The table contains encyclopaedic information and is unnecessary in this article.
I do not understand how the authors calculated the concentration of 10 % in line 164.
On the one hand, it is good that the authors took such chemically different antioxidants. However, the differences that were observed do not quite correlate with their structure and chemical properties. I believe that and UV-Vis and FTIR-ATR spectroscopy should be included for comparison of the spectra of the mentioned model compounds. In the case of the FTIR-ATR spectrum, it would be advisable to include interesting sections. It is difficult to draw any firm conclusions about a possible interaction on the basis of the figure (fig.2)..
There are also typos, editorial errors in the work which should be corrected. For example.
In line 33 a sentence started with a lower case letter
In line 142 "The originality of his work..." instead of The originality of this work....
In conclusion, I think the work is interesting, the research done is interesting, generally the interpretation is also factual. However, some issues should be revisited, the selection of the model compounds mentioned should be justified and their structural and chemical properties should be included in the discussion. Certainly, the work would be even more valuable if studies of mechanical properties were carried out.

Round 2
Reviewer 1 Report
Comments and Suggestions for Authors
The authors adressed all indicated issues.
Author Response
see answer to reviewers file

Reviewer 2 Report
Comments and Suggestions for Authors
The author revised based on my comments, I have no further questions.
Reviewer 3 Report
Comments and Suggestions for Authors
The authors have responded to the comments. I believe that the article as it stands is suitable for publication in Antioxidants with minor changes. I still believe that the nomenclature is incorrect, these given names correspond to specific (different) chemical compounds. If they are common names (in my opinion they are not), this should be made clear. On the other hand, it would be advisable to give the correct chemical name according to IUPAC nomenclature.
Editorial corrections should also be made, e.g:
Line 471: Carboxylic group (COO-) should be "Carboxylic group (COO-)
line 526: NH2 should be NH2, same in line 532
